# Factors That Determine the Adoption Intention of Direct Mobile Purchases through Social Media Apps

**Vaggelis Saprikis ***  **and Giorgos Avlogiaris**

Department of Management Science and Technology, University of Western Macedonia, Kila,
50100 Kozani, Greece; aff00084@uowm.gr
* Correspondence: esaprikis@uowm.gr

**Abstract:** In the last few years, a number of social media e-business models including the social networking giants of Facebook, Pinterest and Instagram have offered direct purchase abilities to both their users and the involved enterprises. Hence, individuals can buy directly without having to leave the social media website. At the same time, there is a significant increase in the number of online purchases through mobile devices. To add to this, nowadays, the vast majority of internet users prefer to surf via their smartphone rather than a desktop PC. The aforementioned facts reveal the abilities and potential dynamics of Mobile Social Commerce (MSC), which is considered not only the present but also the future of e-commerce, as well as an area of prosperous academic and managerial concern. In spite of its several extant abilities and its booming future, MSC has been little examined until now. Therefore, this study aims to determine the factors that impact smartphone users' behavioral intention to adopt direct purchases through social media apps in a country where these kinds of m-services are not yet available. In specific, it extends the well-established Unified Theory of Acceptance and Use of Technology (UTAUT) model with the main ICT facilitators (i.e., convenience, reward and security) and inhibitors (i.e., risk and anxiety). The suggested conceptual model aims to increase the understanding on the topic and strengthen the importance of this major type of MSC. Convenience sampling was applied to gather the data and Structural Equation Modeling (SEM) was then performed to investigate the research hypotheses of the proposed conceptual model. The results show that performance expectancy exerts a positive impact on behavioral intention. Furthermore, all ICT facilitators examined do impact significantly on smartphone users' decision to adopt direct mobile purchases through social media apps, whereas anxiety exerts a negative effect.

**Keywords:** mobile social commerce; social media apps; direct m-purchases; adoption intention; UTAUT; ICT inhibitors; ICT facilitators

## 1. Introduction

The advancements in the mobile and networking industry have considerably reformed individuals' buying behavior in the online environment. Nowadays, the vast majority of people worldwide use a smartphone and its internet access abilities to search for a product or a service, become informed about it, and might proceed to buy it via mobile shopping. At the same time, almost 4 billion people take advantage of social media e-platforms and their features, mainly through their mobile apps [1]. On the other hand, 73% of marketers believe that social media marketing has been "somewhat or very effective" for their firm [2]. As a result, social networking sites such as Facebook, Twitter, Instagram, Pinterest and YouTube have received the attention of brands, retailers, creators and marketers intending to attract and engage individuals with their provided goods. Thus, the contemporary lifestyle where individuals use their smartphones to search and use mobile shopping, together with the extensive use of social media apps to reinforce their purchasing experience, is the impetus behind the rise of the latest advancement of e-commerce, Mobile Social Commerce (MSC) [3,4].

MSC, which is characterized as a more customer-centric m-commerce model [5], takes advantage of its mobile features, such as ubiquity, convenience, interactivity, localization, personalization, flexibility and dissemination [6,7]. At the same time, its social characteristics, such as social knowledge formation [8] and communication medium flexibility [9] through online chat, ratings, comments, posts, information sharing and purchase recommendations from others [10,11] offer additional benefits to the involved entities. Therefore, it is greatly welcomed by modern society, where even more shoppers want a more communal and communicating way of online purchases. According to Tan et al. [6], researchers proved that customers who are escorted by relatives and friends have a tendency to spend more time and money on buying goods. Additionally, the vast majority (91%) of social media users mentioned that they resort to online review and other forms of users' generated content prior to a purchase, and almost half of them (46%) base their decision on these comments [12]. To add to this, 91% of all social media users access social channels through mobile devices, and the 80% of total time spent on such e-business models occurs mobile [13]. Thus, the extensive use of social media apps where people can easily interact and exchange information between each other increases the chances for promotions and sales via them. This is one of the main reasons why social media e-platforms have invested a lot in mobile technology and continuously try to improve their apps' interface and functionality with the aim to attract more and more individuals and engage them with MSC [3,14]. Together, innumerable e-business ventures utilize social media apps as enterprises can easily interact with numerous people simultaneously without the need to spend a fortune [15]. Thus, it goes without saying that contemporary marketing and e-shopping are greatly based on mobile social media apps, where firms can easily interact with customers, customize their actions to every single user, enhance customers' shopping experience, stimulate user engagement and promote sales [16,17].

Based on the aforementioned facts, the scope of this study is to investigate individuals' intention towards MSC adoption. In spite of a considerable number of empirical studies from the academic community and the industry in the contexts of m-commerce and s-commerce, there has been fairly a small amount of research that has investigated MSC, and even fewer concerning its pre-adoption stage [14]. To add to this, Liu et al. [18] mentioned that it is not certain that s-commerce empirical studies do have the same results in the mobile environment.

Therefore, this paper targets to examine the impact of key factors that influence smartphone users to adopt MSC focusing on direct purchases that can be conducted in social media apps. In recent years, a considerable number of social media e-platforms including Instagram, Facebook and Pinterest allow enterprises to sell directly to individuals via their app. To add to this, smartphones are the most common mobile devices for surfing online by far [19]. Thus, this paper investigates users' intentions to utilize these services. To be more specific, the study focuses on smartphone users in Greece who have never used mobile shopping directly from such apps, as direct purchases are currently limited to the USA. The results of this study expect to help enterprises better understand users' MSC motives. In specific, the paper's outcome might assist them develop specific customized strategies with the aim to receive mobile sales directly via social media apps and increase their profits through an alternative, but highly acceptable, novel, shopping channel, which is expected to be available soon to markets out of the USA. The paper suggests a holistic conceptual model that enriches Venkatesh et al. [20]. Unified Theory of Acceptance and Use of Technology (UTAUT) with the major ICT facilitators (i.e., convenience, reward and security) and inhibitors (i.e., risk and anxiety). The enhancement of the well-established UTAUT model with five additional determinants is expected to provide a better understanding of the factors that impact users' MSC adoption and strengthen the importance of the findings. As a consequence, the results of the paper are expected to provide key scientific and managerial insights to both the academic community and the practitioners, as well as improve the understanding of this major type of MSC.

The rest of the paper is divided into six sections. Section 2 presents the extant literature review of MSC. Section 3 provides the suggested conceptual model together with the research hypotheses. Then, the study's methodology (Section 4) and data analysis and results (Section 5) are presented. Finally, Section 6 comments on the outcomes, whereas Section 7 presents the theoretical and managerial implications, the limitations of the study and the future research recommendations.

## 2. Mobile Social Commerce Literature Review

Since 2014, when the first researchers defined MSC [21], there have actually been a small number of academics and industry experts who have examined this field, compared to the broad, global, adoption and intense use of smartphones and social media apps. In specific regarding the study of users' behavioral intention towards MSC during the 2014–2019 period, a total number of fifteen empirical studies have been revealed, according to Sun and Xu's [14] systematic literature review (SLR). For example, Hew et al. [3] investigated the impact of brand loyalty on customers to continue using MSC as well as the negative impact of privacy on this type of e-commerce, and Gao and Bai [22] studied individuals' continuance intention towards mobile social networking services. Chang et al. [23] undertook a quantitative study to investigate what influences users' trust in travel advice through smartphone social media apps, and Han and Park [24] examined the impact of technology readiness on individuals' perceptions and use attitude towards MSC. Furthermore, Tan et al. [25] suggested and investigated a framework to comprehend users' intention towards mobile social advertising adoption, and Song and Hollenbeck [26] examined the significance of social presence in mobile texting. Likewise, Ooi et al. [9] explored privacy issues with reference to MSC users, and Pelet and Papadopoulou [27] studied customers' perceptions and behavior on MSC. Apart from Sun and Xu's [14] SLR study, there have also been other research that examined individuals' intention towards MSC. For example, Chen et al. [28] proved that impulsive buys are controlled by the emotional trust of the recommender and empathy towards the proposed product in the WeChat app. Similarly, Liu et al. [18] proved that perceived usefulness, trust, subjective norm and social support do have an effect on MSC purchase intention.

To our knowledge, however, based on the current literature review and the empirical studies presented, there have been only six pieces of research that examined users' pre-adoption stage in the context of MSC. In detail, Liébana-Cabanillas et al. [4] proved that subjective norms, perceived usefulness and attitude exert a direct effect on users' MSC adoption, while perceived ease of use does impact indirectly. The same year, 2014, Liébana-Cabanillas et al. [29] proved the indirect effect of trust, perceived ease of use and perceived usefulness as well as the direct impact of social influence, subjective norms and attitude in the m-payment adoption in the context of social networks. Likewise, Williams [30] confirmed that perceived innovativeness together with perceived usefulness impact on mobile social media payments, and Hew et al. [31] investigated the main innovation opposition barriers and privacy worries towards MSC adoption intention. Finally, Baabdullah [32,33], in two different studies, utilized an extended version of the UTAT2 model and proved that all the examined factors, except for habit, do exert a statistically significant impact on mobile social network games. Thus, it can be assumed that there is a noteworthy research gap in the pre-adoption stage of MSC. Based on the above mentioned facts, this study aims to contribute to this fairly limited research activity and provide tangible information not only to the academic community but also to practitioners. Particularly, it is intended to a have a renowned model extended (i.e., UTAUT) with the major facilitators (i.e., convenience, reward and security) and inhibitors (i.e., risk and anxiety) of ICT adoption with the objective of improving the understanding of this highly acceptable and somewhat new type of MSC.

### 3. Research Hypotheses and Conceptual Model

Based on the literature review, a proposed conceptual model aimed at exploring the pre-adopt stage of users' behavioral intention towards MSC adoption along with the initial research hypotheses were formulated (Figure 1). The suggested framework extends the UTAUT model with the basic facilitators and inhibitors of ICT. In the rest of this section, these variables are described in detail, and the research hypotheses are developed.

**Figure 1.** Hypothesized research model.

*3.1. UTAUT Variables*

3.1.1. Performance Expectancy

Venkatesh et al. [34] (p. 159), first defined performance expectancy as "the degree to which using a technology will provide benefits to consumers in performing certain activities". Regarding the extant literature review in the contexts of m-commerce and s-commerce, a considerable amount of research verified that it exerts a positive influence on behavioral intention (e.g., [35–38]). Furthermore, performance expectancy is regarded as the strongest UTAUT determinant [20] and also has the highest impact on individuals' adoption intention in a large number of technological innovation empirical studies as well [39,40]. Therefore, it is anticipated that individuals will purchase directly through mobile social media apps if they consider that they will have positive outcomes.

**Hypothesis 1 (H1).** *Performance expectancy positively influences the behavioral intention to adopt MSC.*

### 3.1.2. Effort Expectancy

Effort expectancy is the other fundamental factor of the UTAUT model and is described as "the degree of ease related with the use of the technology" [20] (p. 159). Consequently, as soon as a person perceives that a technology is easy to use and the interaction with the technology is clear and comprehensible, there are more chances for an individual to demonstrate an intention to adopt it [39]. Similarly to performance expectancy, there have been various studies that have confirmed the impact of effort expectancy on adoption intention in m-commerce (e.g., [41,42]) and s-commerce (e.g., [38,43]). Therefore, with reference to this paper, when an individual believes that direct mobile purchases through social media apps are easy to be conducted, there are more chances to take advantage of them. To add to these, previous studies in both m- and s-commerce confirmed the positive influence of effort expectancy on performance expectancy (e.g., [40,43]). Hence, it is assumed that:

**Hypothesis 2 (H2).** *Effort expectancy has a positive effect on (a) performance expectancy and (b) behavioral intention to adopt MSC.*

### 3.1.3. Facilitating Conditions

Facilitating conditions is the third basic construct of the UTAUT model. According to its developers [20] (p. 453), the facilitation conditions factor is defined as "the degree to which an individual believes that an organizational and technical infrastructure exists to support the use of the technology". With reference to this paper, it can be stated that if mobile users have a proper smartphone and are also aware of the steps required to purchase directly through social media apps, their behavioral intention towards this type of MSC will be increased. The original UTAUT model depicts that facilitating conditions do not exert a positive effect on behavioral adoption intention. Up to now, though, there have been various empirical studies that have confirmed the opposite (e.g., [35,42]). To add to this, a noteworthy number of studies have also confirmed the positive, direct effect of facilitating conditions on effort expectancy (e.g., [40,44]). Thus, it is assumed that:

**Hypothesis 3 (H3).** *Facilitating conditions positively impact on (a) effort expectancy and (b) behavioral intention to adopt MSC.*

### 3.1.4. Social Influence

Social influence comprises the last determinant of the original UTAUT and is defined as "the degree to which an individual perceives that significant others believe, such as family and friends, that he/ she should use a particular technology" [20] (p. 451). In line with these researchers, social influence also exerts a positive effect on behavioral intention [20]. This impact has been confirmed by several empirical studies in ICT since peers', family members' and friends' points of view exert a positive impact on users' behavior. In the context of MSC, Liu et al. [18] and Zhang and Wang [45] proved its effect on adoption intention. Furthermore, various studies also confirmed the effect of social influence on performance expectancy. For instance, Khalilzadeh et al. [36] proved that social influence exerts a direct impact on performance expectancy in their m-commerce empirical study. Thus, it is assumed that:

**Hypothesis 4 (H4).** *Social influence positively impacts on (a) performance expectancy and (b) behavioral intention to adopt MSC.*

### 3.2. ICT Inhibitors

3.2.1. Risk

Risk is considered as a vital determinant in the adoption or non-adoption of a technological innovation. According to Forsythe and Shi [46], risk is an anticipated and undesirable situation because it can greatly influence individuals, not only to the level of ICT use but also to the preliminary adoption of a technology. Thus it is regarded as a key ICT inhibitor. As a result, there are a number of empirical studies in the extant literature of MSC where risk's impact was investigated. To be more specific, Hew et al. [3,31] and Liébana-Cabanillas et al. [4,29] confirmed its undesirable effect on MSC adoption intention. On the other hand, Corbitt et al. [47] stated that there should be a link between risk and anxiety as users' anxiety in online shopping can be minimized provided that the perceived risk levels are as low as possible. Based on the aforementioned facts, it is anticipated that there is a negative relationship between risk and MSC adoption intention and a positive connection between risk and anxiety.

**Hypothesis 5 (H5).** *Risk exerts a positive impact on (a) anxiety and a negative impact on (b) behavioral intention to adopt MSC.*

3.2.2. Anxiety

Anxiety is another vital inhibitor to the adoption of a technology. Indeed, it is a situation where individuals feel nervous, uncomfortable and/or aversive at the prospect of using a technology [48]. Several researchers have already studied and confirmed its negative effect on both m- and s-commerce. In specific, Lu and Su [49] and Saprikis et al. [50] confirmed that anxiety exerts a negative effect on m-commerce adoption intention, whereas Saprikis [51] proved its negative effect in the context of s-commerce. However, it should be emphasized that the absence of temporal and spatial restrictions might reveal higher levels of perceived anxiety to individuals in the m-commerce compared to other methods of shopping [52,53]. Likewise, Hourahine and Howard [54] emphasized that anxiety is amplified because smartphone users may lose not only financial but also personal information when they purchase mobile. Thus, it is assumed that the more anxious about MSC the individuals are, the less likely they are to purchase directly through social media apps.

**Hypothesis 6 (H6).** *Anxiety has a negative impact on behavioral intention to adopt MSC.*

### 3.3. ICT Facilitators

3.3.1. Convenience

A vast majority of empirical studies have already examined the effect of convenience on users' consumer behavior to both offline and online environments [55]. Convenience is also regarded as a major factor in the context of marketing [56]. In particular, De Kerviler et al. [57] mentioned the utilitarian value of convenience, while Kim et al. [55] stated that convenience offers not only time but also space utilities; both of them are regarded as vital elements in the mobile environment. Xu and Gutiérrez [58] confirmed that convenience can also be considered as a key facilitator in the context of m-commerce. To add to this, Shankar and Rishi [59] and Xu et al. [60] proved its positive effect on m-banking and tourism mobile apps adoption in correspondence. With regard to the MSC field, Williams [30] proved that convenience exerts a positive effect on m-payment adoption intention on social media e-platforms. Thus, it is expected that the higher the levels of perceived convenience of this type of MSC activity, the greater the adoption rates from smartphone users.

**Hypothesis 7 (H7).** *Convenience exerts a positive impact on behavioral intention to adopt MSC.*

### 3.3.2. Reward

Morgan [61] mentioned that distinctive competencies are fundamental determinants for an individual to feel devoted to a firm and its products. The ubiquity and personalization features of mobile technology along with the flexibility in the communication medium of social media provide great opportunities for enterprises to lure consumers in various ways. Androulidakis and Androulidakis [62] stated that if individuals are subject to being rewarded when they transact via a mobile device, they would be motivated to utilize these services more often. Zarmpou et al. [63] confirmed that reward can greatly impact m-commerce adoption, while Saprikis [51] proved its effect on s-commerce adoption intention. In the same way, Jang et al. [64] highlighted the great effect of discount coupons on continued utilization of s-commerce websites. Therefore, loyalty points, special offer notifications or a limited time discount are expected to convince even more smartphone users to adopt the direct e-purchase abilities of social media apps.

**Hypothesis 8 (H8).** *Reward exerts a positive impact on behavioral intention to adopt MSC.*

### 3.3.3. Security

Launching mechanisms are vital for social media networks as they should guarantee the security of individuals' information and transactions, generate confidence, thus improving attitudes towards them [65]. Especially in the pre-adoption stage where smartphone users do not have any previous experience, lack of such measures can greatly prevent them from adopting MSC. In the same way, Salisbury et al. [66] mentioned that it is paramount for users to feel secure when they conduct financial transactions in the mobile environment as their concerns are minimized. Up to now, various empirical studies proved the effect of security on individuals' intention to adopt a technology (e.g., [67,68]). Likewise, Oliveira et al. [37] and Saprikis [51] confirmed the positive impact of security on m-payment and s-commerce adoption intention in correspondence. In this study, 'security' refers to 'the degree to which an individual believes that mobile social media apps provide secure mechanisms for protecting direct purchases through them'. Thus, it is alleged that the greater the users' security perceptions are, the higher their intention to adopt this type of MSC.

**Hypothesis 9 (H9).** *Security exerts a positive impact on behavioral intention to adopt MSC.*

## 4. Research Methodology

In this section, the literature review, which was adapted to develop the items of the measurement instrument, together with the applied research methodology steps, are analyzed. Then, the data collection procedure along with the demographic characteristics of the sample is presented.

### 4.1. Development of the Measurement Instrument

To investigate the theoretical determinants of the aforementioned suggested conceptual model (Figure 1), a survey was conducted in Greece. A questionnaire was developed based on the extant literature review (Appendix A). Specifically, the basic measurement items to investigate UTAUT constructs (i.e., performance expectancy, effort expectancy, facilitating conditions, social influence and behavioral intention) were adapted from Venkatesh et al.'s [20] study. Concerning the applied ICT inhibitors, the studies of Jarvenpaa et al. [69] and Wakefield and Whitten [70] were adapted to examine the measurement items of risk, and the researches of Compeau et al. [71], Thatcher and Perrewe [72] and Venkatesh and Bala [73] were adapted to investigate the measurement items of anxiety. With regard to ICT facilitators, the empirical study of Kim et al. [55] was adapted to examine the measurement items of convenience; Saprikis et al. [50] and Zarmpou et al.'s [63] researches wre adapted for reward, and Salisbury et al.'s [66] study was utilized to investigate the measurement items of security. It should be emphasized, though, that each

item was measured on a five-point Likert scale. Furthermore, five demographic questions (i.e., sex, age, place of residence, occupation and education) were also included in the measurement instrument.

The questionnaire was first developed in English and tested for content validity by a university English professor. Next, it was translated to Greek language by the same academic because it was intended to be administered in Greece. To examine the measurement instrument and guarantee its consistency, the questionnaire was pre-tested with a sample of 25 respondents in September 2020. The results from the pilot study certified that the measurement items were valid and reliable. Furthermore, to avoid results' skewing, the pilot test data were excluded from the data of the final sample of the study. Convenience sampling was applied targeting individuals who use smartphones and have social network profile(s).

### 4.2. Data Collection

One-thousand four-hundred and thirty-five (1435) Greek internet users were contacted via their social media accounts in October 2020. The e-message included a text explaining in detail the aim of the study and was accompanied by a hyperlink to the e-questionnaire. Five-hundred and sixty-five (565) individuals completely answered the measurement instrument with a response rate of 39.4%, which is rational for this kind of empirical study [37]. The results present that 56.1% of the respondents were female and almost half of them (45.1%) were aged from 18 up to 24 years old. With regard to the place of residence, the vast majority of the research sample live in a town (44.8%) or in a village/countryside (25.4%). Concerning their occupation, the majority of them were students (41.2%), private employees (22.5%) or public servants (18.6%). Finally, more than half of the respondents have attended an undergraduate program in a university or college (57.8%). The detailed demographics of the research sample are depicted in Appendix B.

### 5. Data Analysis and Results

Section 5 presents the reliability analysis of the measurement items, the convergent validity and the discriminant validity between the latent constructs. The section concludes with the discussion on the structural model results and research hypotheses. It should be mentioned, though, that the data of this study were analyzed using IBM SPSS Amos version 24 software.

### 5.1. Reliability and Validity Analysis of the Measurement Items

To measure the reliability of the questionnaire's items, Cronbach's alpha test was applied. The results exceeded the 0.7 threshold [74], ranging from 0.784 to 0.915 (Table 1). Furthermore, a Confirmatory Factor Analysis (CFA) was implemented with the aim of measuring convergent and discriminant validity. The values of all factor loading indicators ranged from 0.523 to 0.89, surpassed 0.4 threshold [75], and were higher than their cross-loadings on the other constructs. Composite Reliability (CR) exceeded 0.6 in all cases [76] (0.787–0.882) and the Average Variance Extracted (AVE) of each construct was above the 0.5 threshold as they ranged from 0.521 to 0.765 [77] (Table 1). To add to this, a comparison between the possible relationships between constructs with the square roots of AVE values was performed (Table 2). The results depict that the square roots of AVE values were greater than the inter-construct correlations [77]. Thus, the aforementioned results revealed that convergent and discriminant validity were both sustained.

**Table 1.** Reliability and convergent validity.

| Factors | Item | Loading | CR | AVE | Cronbach's $\alpha$ |
|---|---|---|---|---|---|
| Performance Expectancy (PE) | 1 | 0.807 | 0.787 | 0.555 | 0.833 |
| | 2 | 0.780 | | | |
| | 3 | 0.637 | | | |
| Effort Expectancy (EFE) | 1 | 0.784 | 0.870 | 0.580 | 0.864 |
| | 2 | 0.780 | | | |
| | 3 | 0.735 | | | |
| | 4 | 0.746 | | | |
| Social Influence (SOC) | 1 | 0.890 | 0.867 | 0.765 | 0.849 |
| | 2 | 0.859 | | | |
| Facilitating Conditions (FAC) | 1 | 0.834 | 0.831 | 0.623 | 0.784 |
| | 2 | 0.719 | | | |
| | 3 | 0.810 | | | |
| Behavioral Intention (BI) | 1 | 0.739 | 0.793 | 0.562 | 0.909 |
| | 2 | 0.792 | | | |
| | 3 | 0.716 | | | |
| Risk (RIS) | 1 | 0.646 | 0.812 | 0.521 | 0.802 |
| | 2 | 0.774 | | | |
| | 3 | 0.788 | | | |
| | 4 | 0.669 | | | |
| Anxiety (ANX) | 1 | 0.770 | 0.873 | 0.584 | 0.858 |
| | 2 | 0.831 | | | |
| | 3 | 0.839 | | | |
| | 4 | 0.811 | | | |
| | 5 | 0.523 | | | |
| Convenience (CONV) | 1 | 0.713 | 0.837 | 0.632 | 0.893 |
| | 2 | 0.820 | | | |
| | 3 | 0.846 | | | |
| Reward (REW) | 1 | 0.776 | 0.812 | 0.590 | 0.880 |
| | 2 | 0.790 | | | |
| | 3 | 0.737 | | | |
| Security (SEC) | 1 | 0.759 | 0.882 | 0.600 | 0.915 |
| | 2 | 0.752 | | | |
| | 3 | 0.811 | | | |
| | 4 | 0.786 | | | |
| | 5 | 0.763 | | | |

**Table 2.** Inter-correlations and square roots of AVE.

|  | PE | EFE | SOC | FAC | BI | RIS | ANX | CONV | REW | SEC |
|---|---|---|---|---|---|---|---|---|---|---|
| **PE** | 0.745 | | | | | | | | | |
| **EFE** | 0.578 | 0.762 | | | | | | | | |
| **SOC** | 0.437 | 0.142 | 0.875 | | | | | | | |
| **FAC** | 0.285 | 0.596 | 0.025 | 0.789 | | | | | | |
| **BI** | 0.671 | 0.498 | 0.383 | 0.316 | 0.750 | | | | | |
| **RIS** | −0.103 | −0.041 | −0.009 | −0.042 | −0.167 | 0.722 | | | | |
| **ANX** | −0.142 | −0.238 | 0.099 | −0.238 | −0.280 | 0.617 | 0.764 | | | |
| **CONV** | 0.601 | 0.579 | 0.318 | 0.351 | 0.600 | −0.098 | −0.232 | 0.795 | | |
| **REW** | 0.569 | 0.459 | 0.363 | 0.353 | 0.642 | −0.162 | −0.196 | 0.532 | 0.768 | |
| **SEC** | 0.565 | 0.454 | 0.399 | 0.339 | 0.627 | −0.460 | −0.275 | 0.468 | 0.643 | 0.775 |

*5.2. Structural Model Evaluation*

Afterwards, Structural Equation Modeling (SEM) was performed to investigate the research hypotheses of the proposed conceptual model. SEM is regarded as a suitable method for investigating simultaneously the underlying hypothesized structural relations between multiple independent and dependent variables. Based on the fact that there was an absence of variables' normality and the sample's small number of cases, the maximum likelihood estimation method was preferred instead of weighted or generalized least squares [75].

The results of the structural model depicted a good model fit. In particular, the measures of goodness-of-fit are above the suggested thresholds in all cases [78–81] (Table 3).

**Table 3.** Summary of model's goodness-of-fit.

| Fit Indices | Cut-Off Point | Model |
|---|---|---|
| $\chi^2/df$ | ≤5.00 | 1.610 |
| Goodness of Fit Index (GFI) | ≥0.90 | 0.914 |
| Adjusted Goodness of Fit Index (AGFI) | ≥0.80 | 0.896 |
| Comparative Fit Index (CFI) | ≥0.90 | 0.970 |
| Normed Fit Index (NFI) | ≥0.90 | 0.925 |
| Incremental Fit Index (IFI) | ≥0.90 | 0.971 |
| Tucker–Lewis Index (TLI) | ≥0.90 | 0.966 |
| Root–Mean–Square Error of Approximation (RMSEA) [90%CI] | ≤0.05 | 0.035 [0.031–0.040] |

The outcome of the hypotheses testing is presented in Table 4 and pictured in Figure 2. As it is clearly depicted, nine out of the thirteen hypotheses were confirmed. The exceptions are the impact of effort expectancy, facilitating conditions, social influence and risk on the dependent variable (i.e., behavioral intention). Thus, H2b, H3b, H4b and H5b are unsupported in this study. On the other hand, performance expectancy shows a positive relationship with behavioral intention ($\beta = 0.33$, $p < 0.001$), effort expectancy exerts a positive, strong effect on performance expectancy ($\beta = 0.58$, $p < 0.001$) and the facilitating condition construct reveals a positive effect on effort expectancy ($\beta = 0.94$, $p < 0.001$). Therefore, hypotheses H1, H2a and H3a are all confirmed. To add to this, social influence positively impacts on performance expectancy ($\beta = 0.42$, $p < 0.001$); thus, H4a is also confirmed.

**Table 4.** Outcome of hypotheses testing.

| Hypotheses | Paths | Path Coefficients |
|:---:|:---:|:---:|
| H1 | PE -> BI | 0.33 *** |
| H2 | (a) EFE -> PE <br> (b) EFE -> BI | (a) 0.58 *** <br> (b) insignificant |
| H3 | (a) FAC -> EFE <br> (b) FAC -> BI | (a) 0.94 *** <br> (b) insignificant |
| H4 | (a) SOC -> PE <br> (b) SOC -> BI | (a) 0.42 *** <br> (b) insignificant |
| H5 | (a) RIS -> ANX <br> (b) RIS -> BI | (a) 0.48 *** <br> (b) insignificant |
| H6 | ANX -> BI | −0.11 ** |
| H7 | CONV -> BI | 0.18 *** |
| H8 | REW -> BI | 0.24 *** |
| H9 | SEC -> BI | 0.21 *** |

** $p < 0.01$. *** $p < 0.001$.

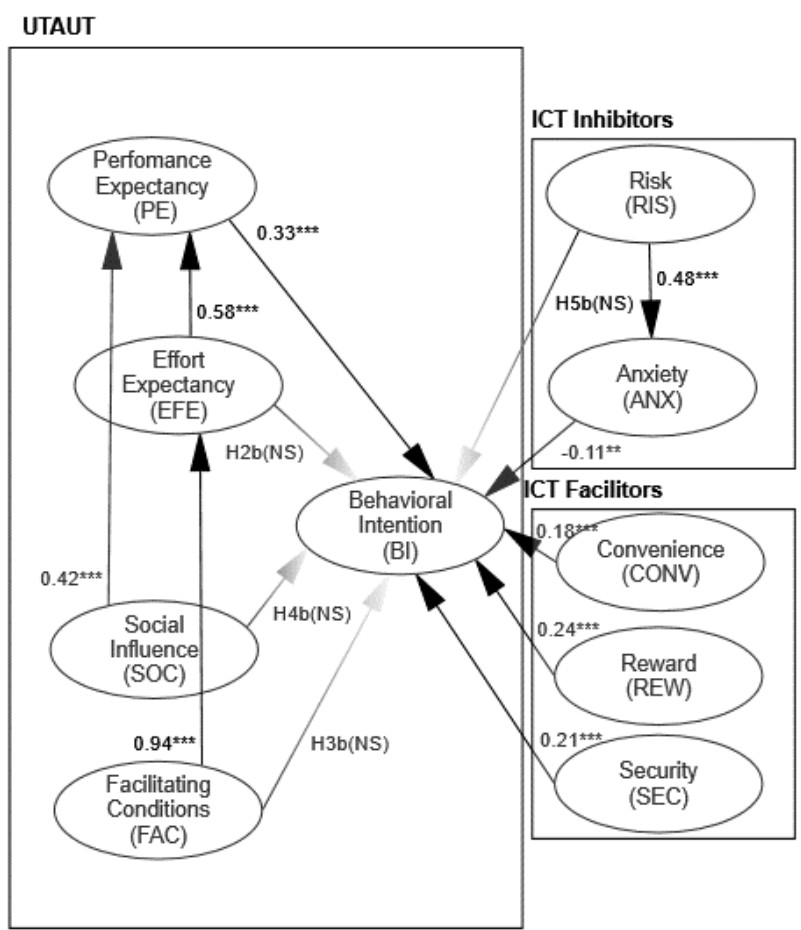

**Figure 2.** Results of the structural model ** $p < 0.01$. *** $p < 0.001$.

With regard to anxiety, as was anticipated, it exerts a negative impact on behavioral adoption intention ($\beta = -0.11$, $p < 0.001$), while risk is greatly associated with anxiety ($\beta = 0.48$, $p < 0.001$). Thus, both H6 and H5a are confirmed. Ultimately, it is considered of great importance that all the ICT facilitators of the model exert a significant positive impact on behavioral intention. In specific, convenience ($\beta = 0.18$, $p < 0.001$), reward

(β = 0.24, *p* < 0.001) and security (β = 0.21, *p* < 0.001) are associated with behavioral intention. Therefore, H7, H8 and H9 are supported in this study. Overall, 75.251% of the variance is mutually explained by the constructs of the structural model.

## 6. Discussion

This paper aims to determine the factors that impact smartphone users' to adopt direct mobile purchases through social media apps. Particularly, it focuses on a comparatively new service of social media websites, where their users have the opportunity to shop directly from them via their mobile apps. The study developed and examined a comprehensive conceptual model, which extended Venkatesh et al.'s [20] UTAUT model with the major ICT facilitators (i.e., convenience, reward and security) and inhibitors (i.e., risk and anxiety). The outcome of the empirical study reveals that all the constructs exert a direct or indirect impact on MSC behavioral adoption intention (Figure 2); however, performance expectancy and reward have the greatest impact on the behavioral intention. Specifically, the results of the structural model depict that nine out of the thirteen research hypotheses were supported (Table 4).

In specific, the findings revealed that three out of the four determinants of the UTAUT were not confirmed from the structural model (H2b, H3b and H4b). This outcome, however, came as no surprise. The fact that the original UTAUT was extended with five more factors seems to alter the involved constructs' dynamics. Actually, this is a normal incident when the SEM method is used and a number of structural relationships between multiple independent and dependent variables take place simultaneously. To add to this, the rejections of basic factors of UTAUT, such as effort expectancy, social influence and facilitating conditions, have already been revealed in several empirical studies where SEM was applied. In particular, Khalilzadeh et al. [36] conducted a study in the context of m-commerce where they proposed an extension of the UTAUT. Their study confirmed the rejection of facilitating conditions construct on individuals' intention to adopt m-commerce. Likewise, Oliveira et al. [37] proved the non-significance of both effort expectancy and facilitating conditions determinants in their m-payment study. It should be emphasized, though, that these researchers did also extend the UTAUT. With regard to the non-significance of risk, it might be attributed to the fact that respondents seem to feel that mobile social apps are not risky to use for buying mobile directly via them. To add to this, the vast majority of social media e-ventures that offer this kind of services, such as Facebook, Pinterest and Instagram, are very popular to individuals, and hundreds of millions of people greatly use them daily for several other reasons. Thus, users might perceive that their novel, direct purchase options are not risky to be adopted.

In contrast, with regard to the supported hypotheses of the conceptual model, performance expectancy exerts the strongest impact on behavioral intention (H1). The great effect of performance expectancy is contingent with the findings of Saprikis and Markos' [82] research. Moreover, the impact of performance expectancy on behavioral intention has already been approved in the extant literature review of MSC (e.g., [36,37]). Concerning the impact of anxiety (H6) on the dependent variable, this is not surprising. Indeed, it is a usual situation for an individual to feel apprehensive about conducting mobile monetary transactions, even if these take place through well-known and daily applied social media apps. At present, there have been several studies that have confirmed the negative effect of anxiety on behavioral intention (e.g., [50,54]). Moreover, users' anxiety might be higher in mobile transactions compared to traditional e-commerce trades because of the absence of time-based and geographical limitations [53]. It should be highlighted, though, that anxiety is the weakest confirmed hypothesis of this study.

Concerning the ICT facilitators, all the examined factors exert a positive impact on behavioral intention. Specifically, convenience (H7) is one of the basic elements that characterize mobile transactions [58]. Individuals can buy any time and in any situation [55] through the ease of access to the Internet via their smartphone. Previous studies in the context of m-commerce (e.g., [59,60]) and MSC (e.g., [30]) have also proved the impact of

convenience on behavioral adoption. With regard to reward, it is a factor that can definitely convince potential buyers to engage with a firm or a product. Thus, the confirmation of the reward (H8) construct reveals the importance of loyalty point discounts, limited-time or/and special offer notifications with the aim of luring individuals towards this type of MSC. To add to this, social media can greatly assist involved enterprises to customize their communication and promotions towards every single member. The confirmation of this determinant is contingent with the empirical studies of Saprikis et al. [50] and Zarmpou et al. [63]; and Saprikis [51] and Jang et al. [64] in the contexts of m-commerce and s-commerce adoption intention in correspondence. As far as this is concerned, though, this is the first time a reward construct has been confirmed in the context of MSC. Lastly, the significance of security (H9) is not surprising. Individuals want to feel secure in order to conduct online transactions; thus, social media apps should guarantee the seamless completion of direct purchases through them, especially in the mobile environment where the small screen of smartphones might incommode the whole process. Despite the fact that there has not been any previous study that confirmed the impact of security to MSC, there are studies that proved its impact in the contexts of both m-commerce (e.g., [37]) and s-commerce (e.g., [51]).

With regard to the other confirmed research hypotheses, all of them reveal the dynamics of the factors of the suggested conceptual model regarding the adoption intention of this type of MSC. In specific, the impact of effort expectancy on performance expectancy (H2a), the impact of facilitating conditions on effort expectancy (H3a) and social influence on performance expectancy (H4a) have already been supported in the extant literature review. To be more specific, Saprikis et al. [40] and Abed [43] have confirmed hypothesis H2a in the contexts of s-commerce and m-commerce in correspondence, while Khalilzadeh et al. [36] and Chung et al. [44] have proved hypothesis H3a in the mobile environment. Moreover, hypothesis H4a has also been confirmed by Khalilzadeh et al.'s [36] research. Finally, the positive effect of risk on anxiety is not surprising. Actually this finding confirmed Corbitt et al.'s [47] allegations that risk reduction can smooth user's anxiety to transact online.

Finally, a comparison between the results of this paper with another recent study in Greece [39] that focuses on non-adopters of m-banking adoption reveals significant findings. In specific, both studies show the great importance of performance expectancy to mobile users' adoption intention. On the contrary, risk was proved to be a vital, direct factor in Giovanis et al.'s [39] work, but not in this study. This may be attributed to the fact that mobile users surf a lot on social media; thus, the potential direct e-purchases via such an e-platform are perceived to be guaranteed from the social media dynamics. As a result, they do not feel buying directly mobile via social media to be risky. Furthermore, this study, compared to that of Giovanis et al. [39], does not prove the direct impact of social influence. Perhaps users are so familiar with visiting social media through their smartphones that they do not believe that their friends, colleagues and family can exert a significant impact on their MSC adoption intention.

### 6.1. Theoretical and Managerial Implications

The findings of this research are expected to contribute to the academic community in several ways. First, the suggested conceptual model along with the hypotheses examination can assist researchers to better understand the comparatively slightly examined MSC field and its pre-adoption stage in specific, as a very small number of empirical studies have been conducted so far [14]. To add to this, the conceptual model extends a well-known and highly applied behavioral model (i.e., UTAUT) with the proposed facilitators and inhibitors of ICT for the first time in the context of MSC. Thus, the paper presents a thorough approach of the topic via the suggested model to the academic community. Furthermore, to the best of our knowledge, the confirmation of reward and security determinants has never been supported before in the context of MSC. Thus, these findings might be the primary step for an alternative approach of their impact, and other researchers could utilize them as a very

useful tool for their future studies. To add to this, the outcomes reveal new insights about MSC adoption as the research was conducted in a country where MSC has never explored. Overall, the conceptual model of this study could be used as a basis and a useful guide for researchers who want to investigate this scientific field even further or wish to refine this model and its findings with other determinants and further explore the topic. For instance, the confirmation or non-confirmation of the determinants of this model in other countries with analogous cultural and socio-demographic characteristics is expected to increase the impact of this research. On the other hand, the improvement of the model with additional determinants from academia would definitely provide a more holistic approach in the context of MSC adoption intention.

On the other hand, the outcomes of this research are also considered important for practitioners. Nowadays, a large number of brands and firms use social media to promote and advertise their products and services. A considerable number of studies have revealed the great importance of MSC to contemporary e-commerce. The fact that 54% of social media members search for a product through them [83] and 91% use a mobile device [13], as well as almost seven out of the ten individuals (71%) having a positive experience with a brand on social media, means that it is likely that they would recommend it to peers and family members [13]. This is a small, but representative, number of statistics that depict their enormous current and potential abilities is presented. Therefore, this study employed marketing and selling approaches as it examined a forthcoming MSC service, which is expected to be available soon in countries other than the USA. Brands and retailers could utilize these findings as a guide to prepare their e-strategy and be as ready as possible to benefit to the greatest extent from this alternative promotion and shopping service. Thus, the outcome of this empirical study could boost their efforts and preparations to engage customers from such a huge pool via mobiles. To add to this, the results of this paper may persuade enterprises that infrequently use social media to increase this activity and follow all the actions required to apply this type of MSC shortly. It is important for enterprises to try to follow the current stream of online consumer behavior where individuals significantly base their searching and buying attitude on mobile social media apps. For example, the confirmation of performance expectancy and convenience show that this type of MSC is expected to be enthusiastically adopted by the public. Individuals will not have to move from one app to another, which is somewhat complex in some cases and can be time consuming, and is regarded as an important advantage for them. Moreover, the support of performance expectancy also shows that individuals perceive that this type of MSC would be useful and aid them in purchasing the right goods much more quickly. To add to this, consumers might visit social media more often for their e-purchases instead of the e-shop of a firm as social media can provide many more products and brands to a single e-marketplace. Therefore, companies may need to start thinking and defining how to adjust their e-strategy towards such a 'threat'. Furthermore, the importance of reward definitely reveals the significance of loyalty point discounts as well as special offers and limited-time notifications. Therefore, firms could use these findings as a tool and may start from now to develop and prepare their upcoming e-strategy. Companies which are a step forward would have a competitive advantage in acquiring a greater market share of this upcoming type of MSC. To summarize, recognizing and valuing this paper's findings would certainly offer an advantage to firms and marketing agencies concerning their forthcoming mobile selling tactics and policies on social media apps.

*6.2. Limitations of the Study and Future Research Recommendations*

The outcomes of this study provide useful insights to both academia and managers; however, there are a number of limitations that should be debated and might lead to further research investigation. First, concerning the selected sample, convenience sampling was applied and only individuals who use smartphones, have social network profiles and responded to the e-questionnaire were included. Thus, a more representative sample regarding the Greek population is desired in a future study with the aim of generalizing

the results. Second, it might be interesting to conduct this study in other countries with the same and different characteristics. The potential results of this cross-cultural investigation may provide useful information and a more holistic approach to the topic. Third, the suggested conceptual model could be enhanced with other determinants based on the extant literature in the broader context of e-commerce. The addition of other factors might increase the percentage of the variance of the model and offer a more comprehensive approach of this type of MSC. Fourth, the proposed conceptual model could also be applied to brands and retailers. Such a study is expected to reveal useful insights concerning the other basic, involved entity, which is indispensable for conducting direct purchases through social media apps. Fifth, it might be interesting if the current users of this type of MSC in the USA were examined (post-adoption stage) in addition to their perceptions before (pre-adoption stage) their decision to use mobile shopping directly through social media apps. Following such a procedure, it might be possible to reveal whether their viewpoints about this type of MSC have changed after the actual use of this service.

## 7. Conclusions

Mobile shopping via direct purchases through social media apps is considered a basic type of MSC with prosperous current and future prospects. Despite the fact that this service is currently limited to the USA, its potential abilities are so great that it is expected to be available soon worldwide, provided by the social networking giants, such as Facebook, Pinterest and Instagram. Up to now, there have been very few research studies on the topic compared to its importance to both individuals and managers [14]. Brands and retailers in particular can benefit greatly from this type of MSC as they have the ability to sell directly through an alternative or an extra channel. To add to this, creators and traditional stores that do not offer their products online can easily start online sales without the need to sustain an e-shop. Thus, the present paper aims to help fill this research gap in the context of MSC and presents an empirical study in which the determinants that impact smartphone users' behavioral intention to adopt direct purchases through social media apps are investigated. In specific, a conceptual model that significantly extends the UTAUT with the fundamental ICT inhibitors and facilitators was developed and examined in Greece, where this type of MSC is not available yet. The findings present that performance expectancy, convenience, reward and security exert a positive effect on individuals' adoption intention, whereas anxiety exerts a negative effect. It should be highlighted, though, that the rest of the suggested factors indirectly impact behavioral intention. Overall, the study contributes to the limited extant literature (i.e., [4,29,31–33]) and provides a noteworthy aid to managers who want to prepare their e-strategy and be ready for these upcoming MSC transactions.

**Author Contributions:** Conceptualization, V.S.; methodology, V.S. and G.A.; software, G.A.; validation, G.A.; formal analysis, V.S. and G.A.; investigation, V.S.; resources, V.S. and G.A.; data curation, G.A.; writing—original draft preparation, V.S. and G.A.; writing—review and editing, V.S.; visualization, V.S. and G.A.; supervision, V.S.; project administration, V.S.; funding acquisition, V.S. and G.A. All authors have read and agreed to the published version of the manuscript.

**Funding:** The APC was funded by the University of Western Macedonia, Greece.

**Institutional Review Board Statement:** The study was conducted according to the guidelines of the Declaration of Helsinki, and approved by the Institutional Review Board (or Ethics Committee) of the University of Western Macedonia (2018).

**Informed Consent Statement:** Informed consent was obtained from all subjects involved in the study.

**Conflicts of Interest:** The authors declare no conflict of interest.

# Appendix A

**Table A1.** Measurement Instrument – Questionnaire.

| Research Variables/Constructs | Items | Sources |
|---|---|---|
| Performance Expectancy (PE) | PE1: I think that using social media apps' direct buy abilities through my smartphone would help me accomplish my purchases more quickly | |
| | PE2: I think that using social media apps' direct buy abilities through my smartphone would increase my chances of purchasing what is important to me | |
| | PE3: I suppose that using social media apps' direct buy abilities through my smartphone would be useful | |
| Effort Expectancy (EFE) | EFE1: I think that learning how to buy directly from social media apps through my smartphone would be easy for me | |
| | EFE2: I think that it would be easy for me to buy directly from social media apps through my smartphone | |
| | EFE3: I think that my interaction with social media apps' direct buy abilities through my smartphone would be clear and understandable | |
| | EF4: I think that I would find social media apps' direct buy abilities through my smartphone easy to use | |
| Social Influence (SOC) | SOCI: People who influence my behavior think that I should use social media apps to buy directly through my smartphone | [20] |
| | SOC2: People who are important to me think that I should use social media apps to buy directly through my smartphone | |
| Facilitating Conditions (FAC) | FAC1: I think that I have the proper smartphone to buy directly via social media apps | |
| | FAC2: I think that I have the knowledge necessary to buy directly via social media apps through my smartphone | |
| | FAC3: I think that I could buy directly via social media apps with my current smartphone | |
| Behavioral Intention (BI) | BI1: I intend to buy directly via social media apps through my smartphone in the near future | |
| | BI2: I predict I would buy directly via social media apps through my smartphone in the near future | |
| | BI3: I plan to directly via social media apps through my smartphone in the near future | |
| Risk (RISK) | RIS1: I think that there would be a significant risk in buying directly via social media apps through my smartphone | [69,70] |
| | RIS2: I think that there would be a high potential for financial loss if I buy directly via social media apps through my smartphone | |
| | RIS3: I think that other people could know information about my transactions if I buy directly via social media apps through my smartphone | |
| | RIS4: I think that buying directly via social media apps through my smartphone would be risky | |

**Table A1.** *Cont.*

| Research Variables/Constructs | Items | Sources |
|---|---|---|
| Anxiety (ANX) | ANX1: I would feel apprehensive about buying directly via social media apps through my smartphone | [71–73] |
| | ANX2: Buying directly via social media apps through my smartphone would scare me | |
| | ANX3: Buying directly via social media apps through my smartphone would make me feel nervous | |
| | ANX4: Buying directly via social media apps through my smartphone would make me feel uncomfortable | |
| | ANX5: Buying directly via social media apps through my smartphone would make me anxious | |
| Convenience (CONV) | CONV1: Buying directly via social media apps would be convenient as I usually carry my smartphone | [55] |
| | CONV2: Buying directly via social media apps through my smartphone would be convenient as I can use it anytime | |
| | CONV3: Buying directly via social media apps through my smartphone would be convenient as I can use it in any situation | |
| Reward (REW) | REW1: I would buy directly via social media apps through my smartphone if they provide information on discounts | [50,63] |
| | REW2: I would buy directly via social media apps through my smartphone if they provide information on special offers | |
| | REW3: I would buy directly via social media apps through my smartphone if they provide me with loyalty points and rewards | |
| Security (SEC) | SEC1: I think buying directly via social media apps through my smartphone is secure to send and receive data/ information | [66] |
| | SEC2: I feel secure to buy directly via social media apps through my smartphone | |
| | SEC3: I would feel totally safe providing sensitive information about myself over direct buys via social media apps through my smartphone | |
| | SEC4: I think buying directly via social media apps through my smartphone would be secure | |
| | SEC5: Overall, I believe that buying directly via social media apps through my smartphone is safe to transmit sensitive information | |

## Appendix B

**Table A2.** Demographic characteristics of the sample.

| Demographic Information | Respondents | % |
|---|---|---|
| Sex: | | |
| Male | 248 | 43.9 |
| Female | 317 | 56.1 |

**Table A2.** *Cont.*

| Demographic Information | Respondents | % |
|---|---|---|
| Age: | | |
| 18–24 | 255 | 45.1 |
| 25–34 | 75 | 13.3 |
| 35–44 | 96 | 17.0 |
| 45–54 | 47 | 8.3 |
| >54 | 92 | 16.3 |
| Place of residence: | | |
| City | 80 | 14.2 |
| Town | 253 | 44.8 |
| Small town | 88 | 15.6 |
| Village/Countryside | 144 | 25.4 |
| Occupation: | | |
| Student | 233 | 41.2 |
| Private employee | 127 | 22.5 |
| Public servant | 105 | 18.6 |
| Freelancer | 37 | 6.5 |
| Unemployed | 30 | 5.3 |
| Other | 33 | 5.9 |
| Education: | | |
| Elementary school | 6 | 1.1 |
| High school | 120 | 21.2 |
| University/College | 327 | 57.8 |
| Master/PhD | 112 | 19.9 |

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
