# Peer review of "Factors That Determine the Adoption Intention of Direct Mobile Purchases through Social Media Apps"

_information, doi:10.3390/info12110449_

Round 1
Reviewer 1 Report
An interesting study that extends existing theories to a relatively new context. Discussing in more depth the practical and theoretical implications of the results would highlight the contribution and the novelty of the research and possibly, attract a larger readership. Given the limitations of the study and the relative lack of novel insights, the paper would benefit from extending the discussion to include a comparison between the results of this research and results obtained in eCommerce/mCommerce research drawing on samples of similar demographics.
Author Response
Dear reviewer,
At first we would like to thank you for taking the time to formulate your useful suggestions and comments.
Following your guidelines, the discussion was improved. In specific, the theoretical and managerial implications subsection was enhanced with the aim to attract even a larger readership. To add to this, in the discussion section, the results of this paper were compared and analyzed to a recent study that contacted to a similar socio-demographic sample (non-adopters of m-Commerce –m-banking- in Greece). Commonly to our study, this research applied the UTAUT model as well.
Your sincerely,
The authors
Reviewer 2 Report
This study is interesting and it has theoretical and managerial implications. Nevertheless, there are some missing links in the study. The authors should pay attention to punctuation marks. It is as important as the text. Check my in-text comments and reverse accordingly.

Author Response
Dear reviewer,
At first we would like to thank you for taking the time to formulate your useful suggestions and comments.
Regarding your guidelines, all the suggestions were added in the paper. In specific, the research methodology along with the data analysis were mentioned in brief in the abstract. Furthermore, the reason why the study was focused on smartphones instead of other mobile devices is also mentioned in the Introduction section (4th paragraph). Finally, in 4.1 section (2nd paragraph) the applied sampling method was mentioned, whereas in the Conclusion section the extant literature review was added as well.
Your sincerely,
The authors
This manuscript is a resubmission of an earlier submission. The following is a list of the peer review reports and author responses from that submission.